# Development of a Proton-Enhanced ESI UPLC-MS/MS Method for the Determination of Tetrodotoxin

**DOI:** 10.3390/molecules27248967

**Published:** 2022-12-16

**Authors:** Tong Li, Ruiguo Wang, Peilong Wang

**Affiliations:** Institute of Quality Standards and Testing Technology for Agro-Products, Chinese Academy of Agricultural Sciences, Beijing 100081, China

**Keywords:** tetrodotoxin, proton-enhanced, UPLC-MS/MS, puffer fish

## Abstract

Tetrodotoxin (TTX) is a kind of low-molecular-weight non-protein neurotoxin. It is one of the most potent neurotoxins found in nature, and it is found in puffer fish and various marine biota. The low sensitivity of previous analytical methods limited their application in puffer fish organ samples. This study established a method for the accurate and fast determination of TTX by reversed ultra-performance liquid chromatography coupled with proton-enhanced electron spray ionization–tandem mass spectrometry. The method yields good peak shapes, high sensitivity and low coeluted interferences. The method was successfully applied to determine TTX in puffer fish tissue samples of about 0.2 g.

## 1. Introduction

Seafood has always been considered nutritionally healthy and an important part of a balanced diet. However, over the past few decades, various types of poisoning incidents have been reported, and this has led to growing concerns with regard to the consumption of seafood. Both public health and aquaculture industries have been affected by such marine-toxin-associated poisoning [1,2]. Tetrodotoxin (TTX) is a potent neurotoxin that selectively inhibits voltage-dependent Na+ channels of nerves and skeletal muscles and is found primarily in Tetraodontidae fishes and Charoniasauliae conches. Furthermore, TTX is stable to heat, light and acid, and it is not degraded by conventional cooking processes, such as heating. It represents a major causative agent of food poisoning by natural animal toxins. The occurrence of TTX has mainly been reported in Asian countries and, more specifically, in Japan, with reports of many cases of TTX food poisoning related to the consumption of pufferfish [3,4].

When puffer fish food poisoning occurs, a positive result for poisoning is normally determined based on the patient’s symptoms and the detection of TTX in residual food. Recently, numerous methods for TTX analysis have been published. Several drawbacks should be mentioned. Bioassays are complicated by problems of inaccuracy associated with the determination of death, difficulty of mouse management that hampers emergency responses and animal welfare issues [5]. On the other hand, physicochemical methods with instruments, such as those using high-performance liquid chromatography (HPLC) with a fluorescence detector and gas chromatography–mass spectrometry (GC-MS), have been reported, but these methods are limited by their complicated operation due to the requirement of degradation by strong alkali and trimethylsilyl derivatization [6,7]. Reports using LC-MS/MS are also emerging, as this instrument has become common. However, the detection limit of LC-MS/MS has always been deemed to be not adequate for the quantitative analysis of real samples with trace levels of TTX due to suppression by matrix effects. Therefore, it is of great significance to develop the present method to achieve the analysis of TTX with high sensitivity and selectivity [8,9].

The generation of gas-phase ions by electrospray ionization depends heavily upon the composition of the droplets formed at the ion source. Much work has focused on analyte-response suppression due to its deleterious effect on sensitivity by the coelute component from biological samples, as well as to help understand the dynamics of the electrospray process [10]. Reagent-enhanced ionization has been known as an effective method to lower the suppression of the matrix and to increase the sensitivity of target chemicals. According to previous studies, there is no doubt that post-column infusion with an enhanced ionization reagent is a reasonable way to achieve signal enhancement given its automation and timesaving characteristics [11,12]. Because TTX contains an amine group, abundant protons could largely increase the formation of [TTX + H]^+^ in the ion source. Acid, as the best potential proton-supply reagent, could obviously enhance the abundance of TTX, even at relatively low concentrations. However, a high content of acid could cause damage to the chromatographic column and affect the retention time of the analyte. This critical technical limitation hampers the establishment of a sensitive method for the determination of TTX.

In this study, we established a method for the accurate and fast determination of TTX via ultra-performance liquid chromatography (UPLC) coupled with proton-enhanced electron spray ionization (ESI)–tandem mass spectrometry. The limit of detection was as low as 20 pg/g, and the preliminary analysis of puffer fish tissues showed that TTX was mainly enriched in the ovary.

## 2. Results and Discussion

### 2.1. Optimization of the UPLC-MS/MS Conditions

The UPLC conditions related to the chromatographic column, the mobile-phase composition and the modifier were optimized after the MS/MS conditions. The MRM parameters were optimized by a TTX standard solution with a concentration of 200 ng/mL injected directly into the mass spectrometer. The positive and negative ESI modes were evaluated. The results showed that the precursor ion signal (*m*/*z* = 320) was detected in ESI+ mode, assignable to [TTX + H]^+^, which was consistent with former studies [9,13,14,15]. Then, the MRM transition from mass 320 to mass 302 was used as the quantitative transition for TTX with the transition from mass 320 to mass 162 as the confirming transition, in accordance with the fragmentation patterns reported in former studies as shown in Figure 1 [16,17].

The LC conditions related to the chromatographic column, the mobile-phase composition and the modifier were optimized after the MS/MS conditions. The TTX standard solutions (20 ng/mL) were applied to carry out the optimization of the LC parameters. In the present study, ACQUITY BEH C18 (100 × 2.1 mm, 1.7 μm, Waters), and ACQUITY UPLC BEH Amide (100 × 2.1 mm, 1.7 μm, Waters) employed in previous reports [9,16], were investigated for the separation of TTX, which are the octadecylsilane stationary phase and amide stationary phase, respectively. Mobile phase A and B consisted of 0.1% formic acid and 1 mmol/L ammonium formate in water and acetonitrile, respectively. The mobile-phase elution gradient and flow rate were optimized to obtain a good chromatographic separation effect. It could be found that the ACQUITY UPLC BEH Amide column and a flow rate of 0.3 mL/min with the following gradient had a sufficient separation effect: 0–0.5 min, 90% B; 0.5–2 min, from 90% to 50% B; 2–3.5 min, 50% B; 3.5–4 min, 50% to 90% B; 4–9 min, 90% B. However, the ACQUITY BEH C18 column with the following gradient did not have a reasonable separation effect: 0–2 min, 80% A; 2–5 min, from 80% to 5% A; 5–8 min, 5% A; 8–9 min, 5% to 80% A; 9–10 min, 80%. The auto-sampler temperature and column temperature were finally set to 4 °C and 40 °C, respectively. The separation chromatogram with ACQUITY UPLC BEH Amide of TTX is shown in Figure 2.

### 2.2. Optimization of the Sample Preparation

An aqueous acetic acid solution or acetic acid methanol solution are usually chosen as the extraction solvents in TTX determination because TTX contains a guanidine group that makes it unstable under alkaline conditions [18,19]. In this experiment, we optimized the extraction solvents, including acid selectivity (formic acid, acetic acid), and co-solvents (water, MeOH, and ACN). Moreover, we compared and tested the ability of 0.1%, 0.3% and 1% acid in solutions to extract TTX from puffer fish tissue samples. Table 1 shows that the 1% acetic acid methanol solution resulted in a mean recovery rate of 90.1% for TTX at 50 ng/mL, which provides an acceptable cleaning extract and decreases ion suppression for the LC-MS/MS analysis. Therefore, 1% acetic acid in methanol was chosen as the final extraction solvent. The lipid content (m/m) of most puffer fish liver and ovaries is approximately >40% [20]. Thus, the lipids in the extracts had to be eliminated or reduced in order to reach the sensitivity of the proposed method. Finally, sonication was carried out for 15 min in an ice-water bath, and the cold extracts were immediately centrifuged at 13,000 rpm at 4 °C for 15 min to remove the lipids and precipitate as much of the solids as possible [21].

### 2.3. Proton-Enhanced Ionization

The addition of a reagent in mass spectrometry analysis has been shown to produce more uniform ionization responses and improve selectivity and sensitivity in chemical analyses [10,11,12,22]. In this study, similar proton-enhanced ionization conditions were observed in direct injection MS/MS analysis, and the formation of [M + H]^+^ ions was only observed. Given that formic acid as a proton-supply reagent can offer abundant protons, the infusion of formic acid could efficiently increase the formation of [TTX + H]^+^. Although UPLC-MS/MS requires a longer running time (9 min) than direct MS analysis, chromatographic separation prior MS/MS yields significant quantitative and qualitative advantages. Figure 2 shows that the post-column infusion of formic acid enhanced the analytical sensitivity of TTX (2000 μg/L) 10-fold compared to the results of no infusion of formic acid. However, the high content of acid in the mobile phase of reversed UPLC would have a bad effect on the retention of the column. As shown in Figure 3, when a relatively high level of formic acid was added to the mobile phases with the separation of ACQUITY UPLC BEH Amide, matrix components of puffer fish samples were coeluted with the target analyte and unsatisfactory peak shapes and sensitivities were found for TTX. Hence, a post-column infusion of formic acid was employed by setting the flow state of the instrument to the combine mode to achieve proton-enhanced ionization, and the optimal condition was to add a 0.4% (*v*/*v*) formic acid solution at an infusion rate of 5 μL min^−^^1^.

### 2.4. Validation of the Proposed Method

The investigation of linearity, limit of detection (LOD), limit of quantitation (LOQ), accuracy and the precision and matrix effect were performed to evaluate method reliability. The TTX standards were diluted to make a series of working solutions with a concentration gradient and were determined, respectively, under the optimal instrument conditions. The external TTX standard-solution calibration curve showed good linearity over calibration ranges from 1 to 2000 μg/L (R^2^ = 0.9992) (Figure 4). The wider linear range, compared with former studies [23], was deemed adequate for analysis in puffer fish tissue samples. The calculated LOD and LOQ was as low as 20 pg/g and 67 pg/g, respectively, which were benefits of the proton-enhanced ionization. The LOD and LOQ in the present study were about 10–50 fold lower than that without the post-column infusion of formic acid, which yielded the confirmation of a positive peak with high reliability, as well as the quantification of trace amounts of TTX [9,12]. The matrix effects of various tissues at different concentration levels were almost negative values, arising from the co-extracted matrix components during the sample extraction procedure in our study. Table 2 shows the matrix effect was more significant in the liver and ovary than that in serum. Moreover, the matrix effect was present in a concentration-dependent manner in the liver and ovary. Due to the matrix effect value ranging from −20% to 20%, the conclusion could be drawn that the matrix effect was acceptable. The results suggest that the sample preparation procedure in this study showed high selectivity in fish samples. As shown in Table 3, six replicates of each spiked concentration level (10, 100 and 1000 μg/L) were performed in blank liver, serum and ovary to evaluate the accuracy and precision of the established method. After extraction and LC-MS/MS analysis, recoveries of TTX at each concentration level ranged from 88.5% to 107.3%, respectively. The RSDs of recovery at three concentration levels were mostly below 10%. For the measurement of intra-day and inter-day precision and accuracy, a spiked solution in a serum sample with concentrations of 2, 20 and 200 μg/L was prepared. Five injection replicates were performed every day and repeated for five consecutive days to test intra-day and inter-day precision, respectively. Table 4 shows that the intra-day and inter-day precision of TTX were all below 8% with recovery rates ranging from 89.2 to 104.8%, which indicated the good state of the instrument. The specificity of the method was determined by analyzing six procedural blanks (consisting of only the extracting solvent). All of the blank samples were processed according to the sample pretreatment procedures, and it was confirmed whether the sample pretreatment procedures would induce background interference to influence the analysis of TTX. Figure 3B shows that the chromatographic conditions had sufficient specificity for TTX, while Figure 5 illustrates that no significant procedural interference substances were detected at retention times of TTX.

The proposed method exhibited excellent reliability, even though the internal standard was not utilized. Because the wide linear range (1–2000 ng/mL) ensured the quantitative analysis of real samples with various concentration levels of TTX and R^2^ = 0.9992, the acceptance criteria was met. Moreover, sonication was carried out for 15 min in an ice-water bath, followed by immediate centrifugation at 13,000 rpm at 4 °C for 15 min, which could remove the lipids and precipitate as much of the solids as possible. The matrix effects were within 20%, and the precisions were no more than 10% in different tissues (liver, serum and ovary), which demonstrated that the pretreatment of the method possessed an exhaustive cleanup, even for high-lipid-content tissues. What is more, the good state of the instrument and the stability of the method was ensured by the intra- and inter-day precision. Additionally, the satisfactory recoveries (88.52–107.33%) of the spiked samples with relatively low RSDs showed excellent suitability and reproducibility for the determination of TTX in puffer fish tissues.

### 2.5. Applications

The proposed method was further applied for the analysis of 13 tissues and organs from puffer fish. Three replicates of each tissue were analyzed. The results show that TTX in the gonads of Takifugu obscurus was 1.2 ng/g, which was comparable to that in the gonads of L. spadiceus (1.7 ng/g) and L. cheesemanii (0.7 ng/g), but extremely lower than that in the gonads of L. lunaris (227.7 ng/g). The concentrations of TTX in the intestine was 0.5 ng/g, which was 10-fold lower than that in intestine of L. cheesemanii (8.7 ng/g) and L. lunaris (5.1 ng/g). What is more, the results also showed a high level of concentration of TTX in the gonads (ovary), followed by the liver, spleen and intestine (Table 5), which is consistent with results published previously [24,25]. As demonstrated by our application of this method to real samples, the developed method is an effective and highly selective technique for the routine determination and confirmation of TTX in a variety of puffer fish tissues and puffer-fish-based products.

## 3. Materials and Methods

### 3.1. Chemicals and Materials

Tetrodotoxin standard (99% HPLC) was purchased from Wako Chemicals (Tokyo, Japan). The stock solution of TTX was prepared in MeOH with a concentration of 1000 mg/L and stored in a brown glass bottle at −18 °C. Methanol (MeOH, HPLC grade) and acetonitrile (ACN, HPLC grade) were purchased from Fisher Chemicals (Bridgewater, NJ, USA). Acetic acid, ammonium acetate and formic acid (FA) were bought from Sigma Aldrich (St. Louis, MO, USA). Distilled water was prepared by a Milli-Q Synthesis water-purification system (Millipore, Bedford, MA, USA).

### 3.2. Sample Information and Sample Preparation

Puffer fish (Takifugu obscurus) samples were purchased from an aquatic product market in Qingdao, which were captured by local fishermen in the Yellow Sea. Samples were packed separately in sealing bags with ice and transported to the laboratory as soon as possible. Then, samples were carefully dissected to obtain edible muscle and other tissues and organs (e.g., liver, ovary, spermary, gallbladder, gill, intestine, skin, heart, spleen, swim bladder, serum and eye), which were subsequently homogenized and then stored at −20 °C until further use.

An aliquot of each individual sample (~0.2 g) was precisely weighed and transferred to an Eppendorf tube. After the addition of beads and 1000 μL of methanol with 1% acetic acid, the samples were vortexed for 30 s, homogenized at 55 Hz for 4 min, and sonicated for 15 min in an ice-water bath. The grinding and sonication steps were repeated three times. The samples were centrifuged at 13,000 rpm and 4 °C for 15 min. The clear supernatant was transferred to an Eppendorf tube, then the remaining samples were added in 250 μL methanol with 1% acetic acid. Then, the samples were vortexed for 30 s and sonicated for 15 min in an ice-water bath, followed by centrifugation at 13,000 rpm and 4 °C for 15 min. Both of the supernatants were combined and evaporated to dryness under a gentle stream of nitrogen, and the residual was reconstituted with 100 μL of 50% acetonitrile with 0.1% formic acid, followed by centrifugation at 13,000 rpm and 4 °C for 15 min. An 80 μL aliquot of the clear supernatant was transferred to an auto-sampler vial for UHPLC-MS/MS analysis.

### 3.3. UPLC-MS/MS Analysis

The UPLC separation was carried out using an Agilent 1290 Infinity II series UPLC System (Agilent Technologies), equipped with Waters ACQUITY UPLC BEH Amide (100 × 2.1 mm, 1.7 μm, Waters). The column temperature was set at 40 °C. The auto-sampler temperature was set at 4 °C and the injection volume was 10 μL. The mobile phases, consisting of 0.1% formic acid and 1 mmol/L ammonium formate in water (A) and acetonitrile (B), were used with the gradient elution. The initial conditions were 90% B and held for 0.5 min, then it was ramped to 50% for 2 min, held for 1.5 min to 3.5 min, and then returned to the initial conditions for 4 min, which were then equilibrated for 5 min before the injection of the next sample.

A SCIEX 6500 QTRAP+ triple-quadrupole mass spectrometer (AB Sciex, Framingham, MA, USA), equipped with an Ion Drive Turbo V electrospray-ionization (ESI) interface, was used for assay development. Nitrogen (99.999%) was employed as the nebulizer, for desolvation and as cone gas. The typical ion source parameters were as follows: ion spray voltage = 5000 V; declustering potential: 65 V; gasification temperature = 400 °C; ion source gas 1 = 60 psi; ion source gas 2 = 60 psi; curtain gas = 35 psi. The MS/MS analysis was operated in ESI positive mode with multiple-reaction monitoring (MRM) for TTX detection. The optimized MS/MS parameters and compound-specific MRM transitions were as follows: the precursor ion was 320 and the daughter ions were 302 and 162 with collision energy 25 eV and 40 eV, respectively. Between the optimized MRM transitions for TTX, the 320/302 pair that showed the highest sensitivity and selectivity were selected as the ‘quantifier’ for quantitative monitoring. The additional transition 320/162 acted as the ‘qualifier’ for the purpose of verifying the identity of TTX. In order to increase detection sensitivities for the TTX, a post-column infusion of formic acid was employed by setting the flow-state of the instrument to the combine mode to add a 0.4% (*v*/*v*) formic acid solution at an infusion rate of 5 μL min^−1^. SCIEX Analyst Work Station Software (Version 1.6.3) and Sciex Multi Quant™ 3.0.3 were employed for MRM data acquisition and processing.

### 3.4. Method Validation

Thorough method validation was performed by evaluating the accuracy, precision, linearity, limit of detection (LOD), limit of quantitation (LOQ), intra-day and inter-day accuracy and the precision and matrix effect. Working standard curves (1, 2, 10, 20, 100, 200, 400 and 2000 ng/mL) were prepared daily by diluting the stock solution with MeOH. The LOD and LOQ were calculated based on S/N (ratio of signal to noise) = 3 and 10, respectively, based on the sample at the lowest spiked concentration level. Accuracy and precision were expressed as recoveries and relative standard deviations (RSDs), which were evaluated at three concentration levels (10 ng/mL, 100 ng/mL, 1000 ng/mL) spiked in six replicates of blank liver, serum and ovary samples. The intra- and inter-run precision and accuracy of the assay (n = 5) were determined as RSDs (%) and recoveries (%). Samples spiked with 2, 20 and 200 ng/mL were analyzed. Five replicates at each concentration were processed to estimate the intra- and inter-day precision and accuracy. Matrix effects were calculated in the serum, liver and ovary by comparing the analytical areas for the extracted samples spiked at concentrations of 10, 100 and 1000 ng/mL with solvent standards, following the equation
$$\mathrm{Matrix}\;\mathrm{effect}\left(\%\right)=\left[\left(\frac{\mathrm{peak}\;\mathrm{area}\;\mathrm{of}\;\mathrm{fortified}\;\mathrm{extract}}{\mathrm{peak}\;\mathrm{area}\;\mathrm{of}\;\mathrm{solvent}\;\mathrm{standard}}\right)-1\right]\times 100\%$$

## 4. Conclusions

In summary, a sensitive method of UPLC-ESI-MS/MS analysis combined with proton-enhanced ionization was established for the simultaneous determination of TTX in puffer fish samples. The calculated LOD and LOQ of the proposed method was as low as 20 pg/g and 67 pg/g, respectively, thus facilitating the determinations of trace levels of TTX in small (e.g., 0.2 g) puffer fish tissue and organ samples.

## Figures and Tables

**Figure 1 molecules-27-08967-f001:**
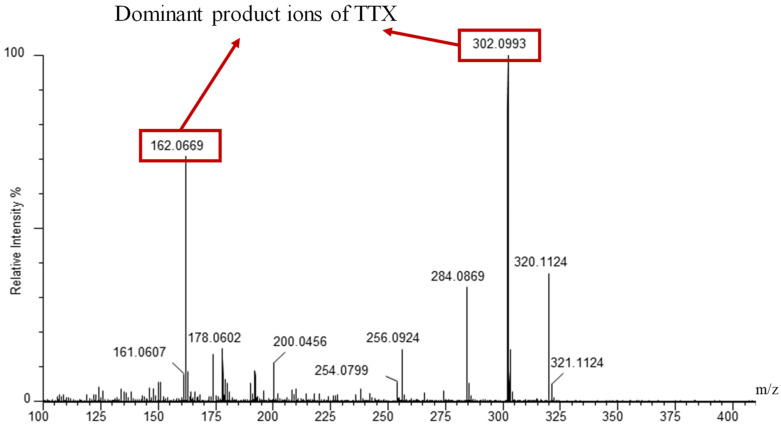
Mass spectra of product ions of TTX in standard solution (2000 μg/L) analyzed by UPLC-QTOF in MS/MS mode with a Waters ACQUITY UPLC BEH Amide column.

**Figure 2 molecules-27-08967-f002:**
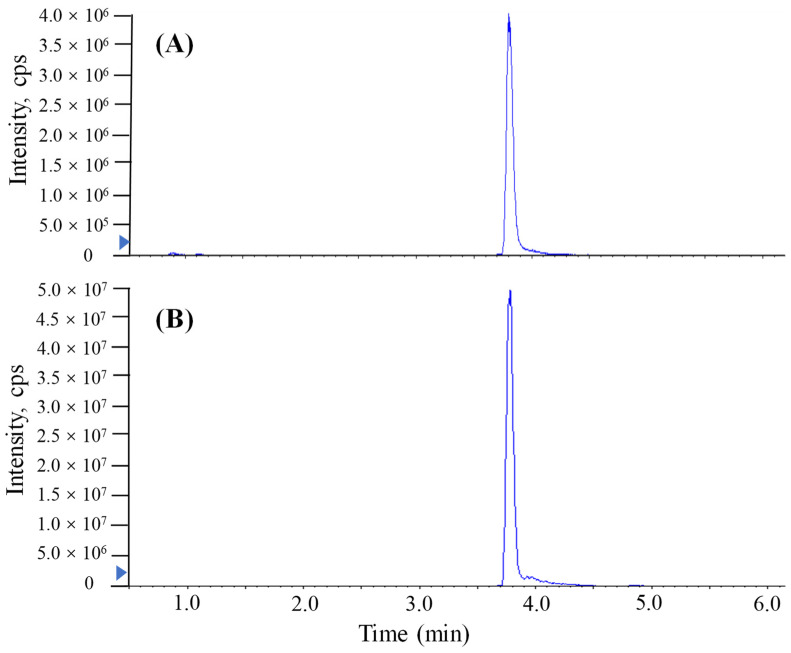
MRM chromatograms of TTX in standard solution (2000 μg/L) analyzed by UPLC-MS/MS with a Waters ACQUITY UPLC BEH Amide column. (**A**) The method conducted without proton-enhanced ionization. (**B**) The method conducted with proton-enhanced ionization.

**Figure 3 molecules-27-08967-f003:**
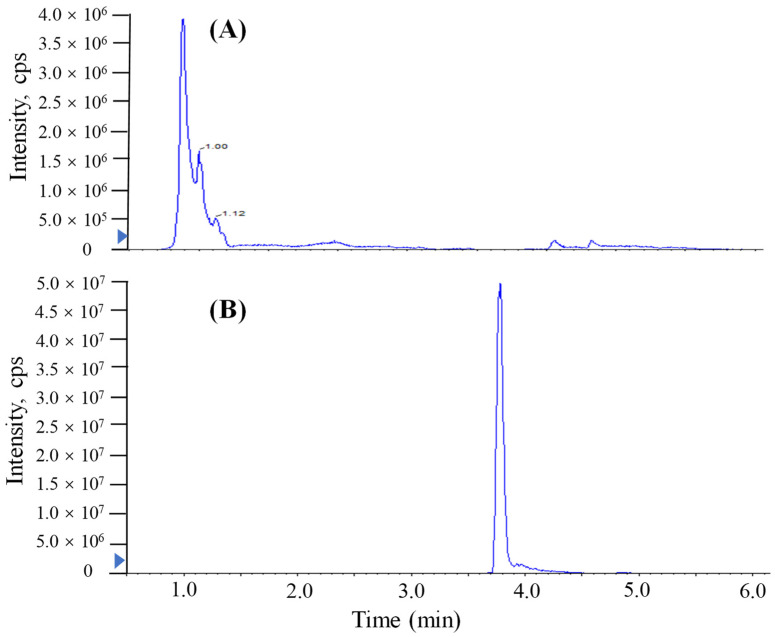
MRM chromatograms of TTX in spiked liver samples (2000 μg/L) analyzed by UPLC-MS/MS with a Waters ACQUITY UPLC BEH Amide column. (**A**) Formic acid added in mobile phase. (**B**) Formic acid injected in the post-column infusion mode.

**Figure 4 molecules-27-08967-f004:**
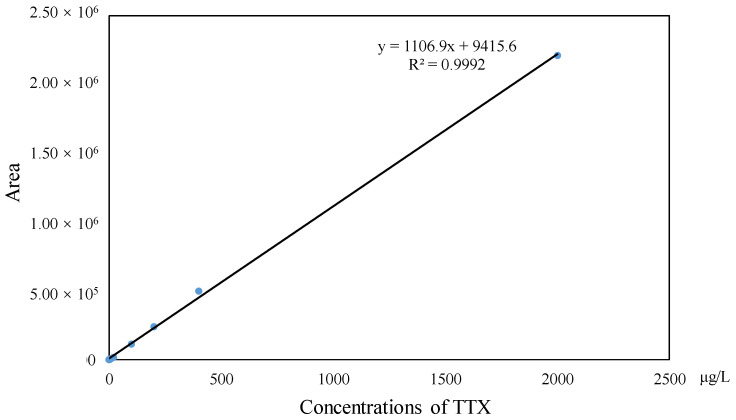
Linear calibration curve of TTX.

**Figure 5 molecules-27-08967-f005:**
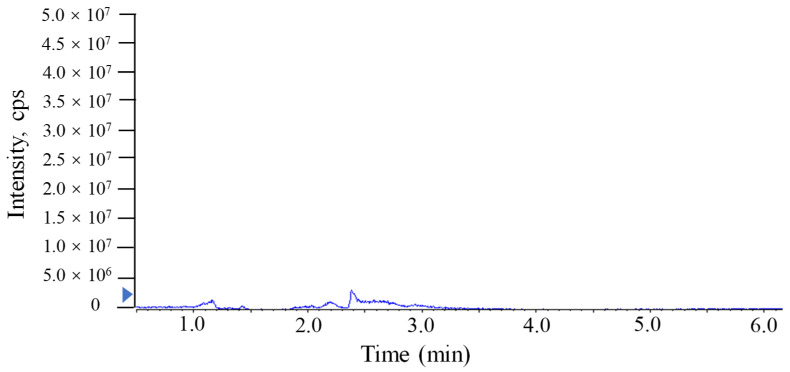
MRM chromatograms of TTX in procedural blank sample analyzed by UPLC-MS/MS with a Waters ACQUITY UPLC BEH Amide column.

**Table 1 molecules-27-08967-t001:** Extraction recoveries of TTX with different conditions, including selectivity of acids and co-solvents. FA means formic acid; AA means acetic acid.

	Aqueous Solution	MeOH Solution	Can Solution
0.1% FA	55.1 ± 7.4%	60.1 ± 4.7%	44.2 ± 2.6%
0.3% FA	61.2 ± 4.8%	55.2 ± 5.1%	52.1 ± 7.5%
1% FA	52.9 ± 1.1%	61.3 ± 10.7%	48.2 ± 7.1%
0.1% AA	68.3 ± 5.2%	78.8 ± 4.2%	68.1 ± 11.8%
0.3% AA	74.5 ± 7.4%	89.2 ± 6.1%	82.5 ± 13.0%
1% AA	77.1 ± 8.2%	90.1 ± 3.4%	74.3 ± 6.7%

**Table 2 molecules-27-08967-t002:** Matrix effects of various tissues at different TTX concentration levels.

Tissue	Matrix Effect % (RSD %) *n* = 6		
	10 μg/L	100 μg/L	1000 μg/L
Serum	−4.21 (3.21)	−3.62 (4.83)	−4.21 (5.72)
Ovary	−14.94 (8.77)	−6.71 (11.56)	−5.32 (9.52)
Liver	−13.34 (11.94)	−5.58 (9.53)	−4.97 (8.80)

**Table 3 molecules-27-08967-t003:** Recoveries and RSDs of TTX, including spiked serum, ovary and liver samples.

Tissue	Recovery (%)			RSD (%)		
	10 μg/L	100 μg/L	1000 μg/L	10 μg/L	100 μg/L	1000 μg/L
Serum	88.52	88.72	97.10	6.23	9.06	7.66
Ovary	89.27	107.33	94.63	8.34	11.29	13.23
Liver	91.20	92.90	94.65	5.66	5.00	4.56

**Table 4 molecules-27-08967-t004:** Inter- and intra-day accuracy and precision.

	Inter-Day Assay		Intra-Day Assay	
	Recoveries (%)	RSDs (%)	Recoveries (%)	RSDs (%)
2 μg/L	89.2	3.5	102.4	5.2
20 μg/L	104.2	4.2	104.8	6.7
200 μg/L	90.5	6.9	94.7	7.7

**Table 5 molecules-27-08967-t005:** Concentrations of TTX in 13 tissues and organs of puffer fish.

Tissues and Organs	Mean (ng/kg)	sd (ng/kg)
ovary	1209.63	76.66
liver	824.23	15.23
spleen	725.60	11.92
intestine	555.08	55.46
serum	549.67	13.98
gallbladder	532.33	5.46
heart	435.80	23.32
swim bladder	100.93	3.19
eye	57.33	13.31
gill	49.20	13.15
spermary	38.27	6.75
muscle	51.83	2.51
skin	48.70	5.65

## Data Availability

Not applicable.

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
