# Peer review of "Development of a Proton-Enhanced ESI UPLC-MS/MS Method for the Determination of Tetrodotoxin"

_molecules, 2022, doi:10.3390/molecules27248967_

Round 1
Reviewer 1 Report
The manuscript is an attempt to develop an LC-MS/MS method for TTX in fish tissues. This I suppose is a useful methodology given that most environmental analysis laboratories do already have the equipment, the method is not cumbersome and the results can easily be validated. I do have some concerns about the manuscript:
1. Proton-enhanced ESI is not new and I am not sure whether is even called like this. Methods of mixing additional acid or other enhancers post-column are not uncommon. MOreover I am not clear as to why the infusion of the 0.4% formic acid is necessary after the column as oppose to just adding it to the mobile phases. That is not explained in the text.
2. In line 31, the term poisoning reason should be replaced with a positive result for poisoning
3. Line 33 should read "several drawbacks should be mentioned"
4. In line 44 it should read "suppression by" and in line 51 it should be "has been"
5. In Line 76 the references 9 and 16 do not have a bracket.
6. There is some confusion on the mobile phase. One of the columns is a C18 and the other one is an Amide HILIC type. Gradients for reverse phase go from aqueous to organic while HILIC columns use gradients going in the opposite direction (reverse-reverse). Lines 79-84 looks like a typical HILIC gradient starting with 90% acetonitrile. Where is the gradient for the reverse phase method?
7. why was acetic acid chosen for extraction of the tetrodoxin and why were other acids such as formic, carbonic not tried. How about co-solvents (MeOH, ACN) etc. ? Did the authors made an assessment on the amounts of contaminants present in the samples by trying different conditions and comparing spectral ion intensities for the different samples?
8. In line 106 the justification for the "proton enhanced" infusion said it had a bad effect on the column separation. Given that no results are shown an explanation is required here. I would assume that TTX would bind to the HILIC column with equal affinity regardless of the formic acid concentration.
9. Line 119 should be were instead of "was". In line 135 it should read "Three" not "3".
10. Figure 1 does not have the cps equivalent or scale on the y-axis. It is hard to compare the two plots. A calibration curve was not included.
11. Table 2 should read "intestine" in line 145.
12. Lines 179-184 have the description of the mobile phases. They appear to be for a HILIC phase, supplemented with NH4HCO2 for both A and B, but this does not correlate well with the previous description. Moreover the experiments should be fully described in the experimental section. Please clarify the conditions for either chromatography (HILIC and C18)
13. In line 207 it should read MeOH, not MetOH. In line 168 it should read "was transferred".
There is no supplementary information provided for the calibrations, spectral data or ion fragmentation structures provided.
Reviewer 2 Report
In this manuscript, a sensitive LC-MS/MS method for quantitative analysis of TTX, a potent neurotoxin, in pufferfish tissues was established and validated. The method was applied to a tissue distribution study of TTX in pufferfish. This manuscript will provide useful information to analysts and biologists in chosing methods for quantification of TTX in biological matrices. However, there are some major issues should be addressed before it is taken into consideration for acception in the molecules:
1) The novelty of this work is not well articulated in the manuscript. LC-MS has been used for the quantitative analysis of TTX in biological tissues. It is important to clearly describe what new science is reported in this study.
2. The selection of internal standard is crucial to the accuracy of biological analysis methods. Is there no internal standard used in this study? Why?
3. The authors used proton-enhanced ionization to improve selectivity and sensitivity, however, no figure or table data was provided in the manuscript, how can we interpret the result?
4. How about the extraction recovery of TTX? No figure or table data was provided in the manuscript.
5. How about the intra-day and inter-day reproducibility and robustness of the method? No figure or table data was provided in the manuscript.
6. Page 3, line 115-117, the authors mentioned: “The external TTX standard solution calibration curves showed good linearity over the calibration ranges from 0.1 to 4000 μg/L (R2 = 0.9992).” however, page 6, line 206-207, the authors mentioned: “……standards curves (1, 2, 10, 20, 100, 200 and 400 ng/ml) were prepared daily……”. They are not consistent. How were the calibration curves produced? The calibration curves should be provided in the manuscript as a figure.
7. The author should compare the level of TTX in pufferfish tissues with that reported in the literatures.
Round 2
Reviewer 1 Report
I appreciate the response of the authors to my and the other reviewer's concern regarding the manuscript. The inclusion of three additional figures certainly helps in making the case. The authors are now showing the gradients used in the 2 columns. I do have a few questions as a result of this:
1. The gradient for the BEH C18 has a 1 min re-equilibration step at the end, equivalent to ~1 column volume. This could be the reason why TTX does not "separate well" due incomplete re-equilibration. How do they know the columns are getting fully re-equilibrated?
2. The legends in Figures 1,2 and 3 are incorrect. There is no mention of the column used in each figure so it is hard to relate to the method. Figure 4 has no axis units.
3. Signal enhancement reagents are usually compounds that are added post column and which are not present in the mobile phase program, infused from a highly concentrated stock. In this case formic acid is at 0.1% in the gradients and 0.4% in the post-column infusion. The flows are 300 and 5 uL/min respectively. Thus the acid concentration in the column efluent after infusion is:
(300*0.1% + 5*0.4%)/305 = 0.105% formic acid with infusion
(300*0.1%)/300 = 0.1% formic acid without infusion
That is an increase of 5% (5 parts in 100) in the proton concentration. How can that increase in protons bring about a 10-fold increase in the signal intensity?
Reviewer 2 Report
More detailed information have been added in the context according to my suggestions, which make the whole paper expressing more rigorously. However, because there are several major drawbacks in the manuscript, it is recommended to rejected.
1. The specificity of the method is key for accurate quantification. How to evaluate the specificity of the proposed method? No data or chart on method specificity is available in the manuscript.
2. I am not satisfied with the explanation of why internal standard is not used. The authors should discuss why internal standard is not used in the manuscript.
3. The authors mentioned “For measurement of intra-day and inter-day precision and accuracy, spiked solution in serum sample with concentration of 2, 20 and 200 ng/mL was prepared.” Why these concentrations was selected for the measurement of intra-day and inter-day precision and accuracy?
In addition, these concentrations were not consistent with the “ As shown in Table, 2, six replicates of each spiked concentration level (10, 100 1000 μg/L) were performed in blank liver, serum and ovary to evaluate the accuracy and precision of established......”(Page 3, Line 146-149).
4. The matrix effect of the method is key for accurate quantification. Dose the matrix effect is more significant in tissues than that in serum. Dose the matrix effect is present in a concentration-dependent manner?
5. For Figure 1. “Product-ions spectrum of TTX by UPLC-QTof in MS/MS mode.”, Why dose the author provide a Q-TOF spectrum since they used MRM for quantitation? What is the concentration of TTX? How can we interpret the product-ions spectrum? The authors should give annotations to the abundant ions.
6. For Figure 2. What is the concentration of TTX? Is it detected in serum, or tissues? More information should provided.
7. For Figure 3. What is the concentration of TTX? Is it detected in serum, or tissues?More information should provided.
8. For Figure 4. Linear calibration curve for TTX. Is it detected in serum, or tissues?More information should provided.
9. Table 3. Inter- and intra-day accuracy and precision. Is it detected in serum, ovary or liver samples. More information should provided.
Author Response
Please see the attachment。
